# Current Consumption of Traditional Cowpea-Based Dishes in South Benin Contributes to at Least 30% of the Recommended Intake of Dietary Fibre, Folate, and Magnesium

**DOI:** 10.3390/nu15061314

**Published:** 2023-03-07

**Authors:** Lorène Akissoé, Youna M. Hemery, Yann E. Madodé, Christèle Icard-Vernière, Isabelle Rochette, Christian Picq, Djidjoho J. Hounhouigan, Claire Mouquet-Rivier

**Affiliations:** 1Qualisud, University of Montpellier, IRD, CIRAD, Institut Agro, Avignon University, Université of La Réunion, 34090 Montpellier, France; 2Laboratoire de Sciences des Aliments, Faculté des Sciences Agronomiques, University of Abomey-Calavi, Abomey Calavi 01 BP 526, Benin

**Keywords:** pulses, food atlas, food frequency questionnaire, traditional dishes, nutritional intakes

## Abstract

Regular consumption of legumes is recommended worldwide for its environmental and health benefits. Cowpea, the most frequently consumed pulse in West African countries, is rich in nutrients and health-promoting bioactive compounds. A one-week retrospective food frequency questionnaire was used to estimate the contribution of the cowpea-based dishes to the recommended nutrient intake (RNI), based on their consumption frequency, intake, and nutritional composition. Participants were 1217 adults (19–65 years) from three urban or rural areas in southern Benin. Out of all respondents, 98% reported that they usually consumed cowpea-based dishes. The mean consumption frequency was 0.1 to 2.4 times/week, depending on the type of cowpea-based dish. The mean amount consumed was 71 g and 58 g of seeds/adult/day in urban and rural areas respectively. The mean daily contribution of cowpea-based dishes to RNI was 15% for energy, 42% for fibre, 37% for magnesium, 30% for folate, 26% for protein, and just above 15% for zinc and potassium. Thus, such regular cowpea consumption should be maintained.

## 1. Introduction

The increasing industrialization and urbanization worldwide are associated with ‘nutrition transition’, characterized by diets richer in fats and sugars, and with more sedentary lifestyles [1]. As a consequence, the prevalence of overweight and obese populations, and related chronic diseases, is constantly increasing [2]. In Benin, the prevalence of diabetes in 2014 and high blood pressure in 2015 were 7% and 28%, respectively, in adults. Moreover, in 2016, obesity (body mass index ≥30) concerned 23% of women and 6% of men [3]. Therefore, strategies are needed to prevent these dietary-related noncommunicable diseases. For instance, the consumption of recommended food groups, such as pulses, should be encouraged due to their health benefits [4]. Pulses (e.g., dry beans, peas, chickpeas, cowpeas) are particularly interesting for fighting against malnutrition [5], because they are rich in key nutrients and in some health-promoting bioactive compounds [6]. Pulses are an important source of dietary proteins with a good essential amino acid profile, complementary to that of cereals. They are rich in minerals (particularly magnesium, potassium, zinc, and possibly also iron) [7] and vitamins (E and B groups) [8]. They have a low lipid content [9], and provide large amounts of dietary fibre and various bioactive compounds that help prevent metabolic disorders. Several studies suggested that the consumption of legumes could help aid weight loss, possibly due to their low lipid and high dietary fibre contents that favour satiation and satiety [6]. Pulse-based foods have low glycaemic index values, recommended for people with type 2 diabetes [10]. Regular pulse consumption helps reduce LDL cholesterol levels and lower blood pressure [11]. Wang et al. [12] showed associations between legume consumption and gut microbiome diversity. 

The ongoing nutritional transition in many urban areas of sub-Saharan Africa, characterized by an increased demand for processed and quick-to-prepare foods, leads to a gradual shift from the traditional diet (high in fibre, low in fat and sugar), to a diet rich in refined foods, low in fibre, and high in fat, salt and sugar [13]. The long time required to prepare cowpea-based dishes and the presence of anti-nutritional factors that cause intestinal disorders could also lead to turn away from traditional cowpea-based dishes in these areas. In addition, their nutritional composition could be affected by processing (as reported for the traditional processing of cowpeas into Ata and Ata-doco, two cowpea-based doughnuts consumed in Benin [14]), leading to a reduction of their nutritional value.

Ten years ago, in Benin, Madode et al. [15] investigated the consumption patterns of ten traditional cowpea-based dishes (among the 18 dishes identified as available in the southern part of the country) and their qualitative consumption frequency (regularly, often, occasionally). They found that Ata (doughnut), Abobo (stew), and Atassi (mixed dish of cowpeas and rice) were the most frequently consumed cowpea-based dishes. They also determined some nutritional traits of these dishes; however, they did not have data on the consumed quantities and could not evaluate their contribution to the recommended nutrient intake (RNI). 

The aims of our study were: (i) to update data on cowpea consumption patterns in South Benin, and (ii) to estimate the nutritional intakes linked to the consumption of cowpea-based foods. Among the 18 cowpea-based dishes listed in the work by Madode et al. [15], the nine that are currently the most popular were selected for this study. These dishes can be classified into three groups: doughnuts (Ata, Ata-doco, Ataclè), mixed dishes (Abla, Atassi, Djongoli), and stews (Adowè, Abobo, Vêyi) (Figure 1 and Appendix A). First, we carried out a food consumption survey among 1217 adults (19–65 years of age) in an urban area (Cotonou) and two rural areas (Adjohoun, Allada) in the southern part of Benin, using a food frequency questionnaire (FFQ) that we validated. Then, we sampled the nine dishes from street food vendors in Cotonou to determine their nutritional composition (proximate composition, minerals, folate, and thiamine). Using the FFQ and nutritional composition data, we determined the contribution of these dishes to the adult RNI of dietary fibre, proteins, magnesium and folate, zinc, potassium, and thiamine.

## 2. Materials and Methods

### 2.1. Cowpea Consumption Survey Using a FFQ

#### 2.1.1. Study Areas and Cluster Sampling

We carried out this study in three different municipalities in the southern part of Benin, Cotonou (an urban area of ~680,000 inhabitants), at the end of 2017, and Adjohoun and Allada (two rural areas of ~75,000 and ~128,000 inhabitants, respectively) in early 2019. In Benin, municipalities encompass administrative units called ‘arrondissements’ that, in turn, contain city districts (for urban municipalities) or villages (for rural municipalities). Adjohoun is a rural municipality in the Ouémé Department, that produces 65% of the total amount of cowpea seeds grown in Benin [16]. Conversely, Allada, a rural municipality of the Atlantique Department, is a low-cowpea production area.

In each study area, we carried out representative two-step random sampling by considering city districts or villages as clusters (Figure 2). In the first sampling step, we selected two or three clusters per arrondissement using random numbers generated by the Excel software. In the second step, we randomly chose a direction by spinning a pen at the centre of each cluster for household selection. One adult, aged between 19 and 65 years, was interviewed in each selected household. An equal number of men and women were selected by cluster. The sample size was chosen based on previous studies that used FFQs in Africa [17]: 641 adults in Cotonou were surveyed and 576 in Adjohoun and Allada (288 respondents per rural area).

#### 2.1.2. Development of the Food Atlas

A food atlas was developed for the nine cowpea-based dishes described in Appendix A and presented in Figure 1, following the guidelines proposed by Nelson and Haraldsdóttir [18]. For each cowpea-based dish, photographs of four portion sizes were included in the food atlas (an example is provided in the Appendix A). The photographs illustrate portions corresponding to the four best-selling prices for each cowpea-based dish, as sold by street vendors. After determining the best-selling prices by recording the consumers’ purchase prices at different selling places, we weighed the portion size for each best-selling price at several selling places and used the mean value as the portion size in the atlas. For each cowpea-based dish, there were nine possible portion size choices, with four presented as photographs (B, D, F, H) and five intermediate portion sizes without photographs (A, C, E, G, and H) (Appendix A). This allowed covering a wide range of cowpea-based dish portions consumed by the participants.

#### 2.1.3. Validation Studies

Food atlas validation

We carried out a validation study in Cotonou to compare the portion sizes chosen by respondents listed in the food atlas to the corresponding weighed records. A convenient sample size of 50 customers per cowpea-based dish produced as street food was initially targeted. Finally, the number of people we actually surveyed varied between 19 and 52 respondents per dish, because for some dishes, there were very few street vendors. The study was performed on two days at various selling places for each cowpea-based dish. On day 1, the quantity of the cowpea-based dish consumed by the respondent was weighed using a kitchen scale (Soehnle, weighing capacity 5 kg, weight accuracy 1 g). An appointment was made to perform the 24 h recall (Figure 2) using the food atlas on the next day (day 2). Respondents were not aware that the portion size consumed the day before would be asked again on day 2.

FFQ validation

We validated the FFQ with a sample of 50 respondents in each of the three study areas. Briefly, three 24 h recalls were carried out in the same week (two recalls during weekdays and one in the weekend), followed by the FFQ at the end of the week (Figure 2). The usual portion size determined with the FFQ data was then compared with the mean estimated weights during the three 24 h recalls.

#### 2.1.4. FFQ Study Description

The FFQ included two parts. In the first part, data on the household head’s socio-professional characteristics, household income and expenses (feeding and clothing), household belongings, and housing characteristics were collected to characterize the household socio-economic index. In the second part, data on the consumption of cowpea-based dishes and the estimation of the quantities consumed over one week were recorded. 

During the FFQ-based interviews, we first asked general questions about the consumption of various legume species (results published elsewhere [19]). This was followed by questions on the cowpea-based dish types consumed by the respondent during the previous week (named 7-day recall, thereafter). For each cowpea-based dish, we recorded how many times it was consumed, the last place of consumption, and the quantity usually consumed (estimated using the food atlas).

#### 2.1.5. Data Collection

Data were collected on digital tablets, using surveyCTO™-generated Excel forms to allow their quick and correct transfer to the SurveyCTO™ platform.

For the FFQ validation study, data were collected on paper forms and then entered in the Epidata software (version 3.1).

### 2.2. Sampling of Cowpea-Based Dishes

The nine most frequently eaten cowpea-based dishes were sampled at 27 street food vendors (i.e., three vendors for each type of dish) in Cotonou and transported in a cooler containing ice to the laboratory, where they were stored at −20 °C until analysis.

### 2.3. Nutritional Composition Analysis

Proximate composition.

Lipids (AOAC Official Method 2003.06) and proteins (NF V03-050, AFNOR, 1970) were determined for each cowpea dish using standard methods [20]. Ash contents were determined by calcination in a furnace at 530 °C. Dry matter contents were determined by oven drying at 105 °C for 24 h. The total dietary fibre (TDF) was determined using an enzymatic–gravimetric method (Megazyme K-TDFR Kit), as described by Njoumi et al. [21]. Available carbohydrates were determined by difference using the following formula: (100 − (Water content + Lipid + Protein + Ashes + TDF)), and the energy value using the Atwater coefficients [22].

Micronutrients.

Minerals were extracted with a closed-vessel microwave digestion system (ETHOS-1, Milestone, Italy) [23]. Extracts were then analysed for total iron, zinc, calcium, potassium and magnesium by optical emission spectrometry, using an ICP-OES 5100 apparatus (Agilent Technologies, Les Ulis, France). Total folate content was analysed by trienzymatic extraction followed by a microbiological assay, using *Lactobacillus rhamnosus* ATCC 7469 as the growth indicator microorganism [24]. Total thiamine content was determined by chromatographic analysis using the Waters Acquity^TM^ Ultra Performance LC (UPLC) system (Waters, Milford, MA, USA), equipped with an Acquity^TM^ fluorescence detector and an Acquity UPLC^TM^ column, using a method adapted from Schmidt et al. [25].

### 2.4. Calculation and Statistical Analyses

#### 2.4.1. Analysis of Data from the Validation Studies

For both validation studies, we used Spearman’s correlation tests after checking the data normality. We assessed the concordance between methods using the Bland–Altman test. For this, we log-transformed data [26] to narrow the limits of agreement. We calculated the antilogarithms of the limits of agreement to obtain the ‘24 h recall over-weighed record’ and ‘usual portion size from the FFQ over the mean value of the three 24 h recalls’ ratios for the food atlas and FFQ validation, respectively. These ratios were then expressed as percentages of agreement [27].

#### 2.4.2. Determination of the Socio-Economic Index

The household socio-economic index (SEI) was determined separately for the rural and urban populations by considering 17 variables related to the housing quality, assets owned, household income and expenses, and household head’s socio-professional characteristics. A multiple-correspondence analysis with the selected variables was performed using the Rstudio software (version 3.5.1). We chose the axis that accounted for the highest percentage of the inertia, and performed an ascending hierarchical classification with the coordinates of the individuals on this axis. The dendrogram partitioning after the ascending hierarchical classification allowed the generating of socio-economic classes. Then, the surveyed households were clustered in three different SEI classes in each area: low, middle, and high (Table 1).

#### 2.4.3. Estimation of the Contribution to the RNI

We determined the contribution of each cowpea-based dish to the RNI using the daily consumed quantity for each traditional dish during the 7-day recall (estimated with the usual portion size) and the daily frequency of consumption (number of consumption times over a week divided by 7). For all respondents, we calculated the daily nutrient intake and compared them to the mean RNI for men and women. We used the reference values provided by FAO and WHO [28,29] for mineral, vitamin and protein requirements, and the values by ANSES [30] for dietary fibre (Appendix A). We divided the daily nutrient intakes (calculated as the amount of nutrients contained in the usual portion multiplied by the daily frequency) by the reference values, to obtain the contribution to the RNI, expressed as a percentage.

#### 2.4.4. Calculation of the Weighting Coefficients for the Statistical Analyses

As the number of city districts/villages or their population density may vary from one arrondissement to another, we calculated a design weight (Dw) that corresponded to the representativeness of each city district/village in its cluster, and applied it to each respondent. The Dw is defined as the inverse of the probability to include a respondent in the sample [13]. In our study, considering the two-step random sampling method used (Figure 2), the Dw corresponded to the product of the inverse of the probabilities at each step. We calculated these probabilities using the following equations:

First step: sampling of city district/rural villages in the arrondissement
(1)$${\mathrm{P1}}_{i}=\frac{\mathrm{Number}\;\mathrm{of}\;\mathrm{city}\;\mathrm{districts}\;\mathrm{or}\;\mathrm{villages}\;\mathrm{selected}\;\mathrm{for}\;\mathrm{the}\;\mathrm{FFQ}}{\mathrm{Total}\;\mathrm{number}\;\mathrm{of}\;\mathrm{city}\;\mathrm{districts}/\mathrm{villages}\;\mathrm{in}\;\mathrm{the}\;\mathrm{arrondissement}}$$

Second step: sampling of subjects within city districts/rural villages
(2)$${\mathrm{P2}}_{i}=\frac{\mathrm{Number}\;\mathrm{of}\;\mathrm{subjects}\;\mathrm{selected}\;\mathrm{in}\;\mathrm{the}\;\mathrm{city}\;\mathrm{district}\;\mathrm{or}\;\mathrm{village}}{\;\mathrm{Eligible}\;\mathrm{adult}\;\mathrm{population}\;\mathrm{of}\;\mathrm{the}\;\mathrm{city}\;\mathrm{district}\;\mathrm{or}\;\mathrm{village}}$$

The formula to determine the design weight (Dw_i_) per respondent (i) was:(3)$${\mathrm{Dw}}_{i}=\frac{1}{P1_{i}*P2_{i}}$$

To perform the statistical analysis with a weighting factor that corresponded to the actual size of the sampled population (n), we calculated a weighed design weight (WeDwi) for each respondent as follows:(4)$${\mathrm{WeDw}}_{i}=\frac{{\mathrm{Dw}}_{i}*n}{\sum_{i=1}^{n^{\mathrm{th}}}{\mathrm{Dw}}_{i}}\;;\;\mathrm{with}\;\sum_{i=1}^{n^{\mathrm{th}}}{\mathrm{WeDw}}_{i}=n$$

The weighting was realized for urban and rural data separately to perform the statistical analyses per area.

#### 2.4.5. Statistical Analyses

Data were analysed with the Rstudio software (version 3.5.1). We used the weighting variables as a correction factor for the statistical analyses with the package ‘Survey’, to ensure that the results were representative of all studied areas. To identify the factors (SEI, sex, age, education level, place of living) that influenced the quantity of cowpea-based dishes consumed, we used general linear models (GLM) adapted to the weighted data and also the one-factor analysis of variance, *t*-test, and chi-2 test. The level of statistical significance was set at *p* < 0.05.

## 3. Results

### 3.1. Validation of the Food Frequency Questionnaire

The validation of the portion sizes in the food atlas, using the 24 h recall data and the recorded weights, gave Spearman’s correlation coefficient values that ranged between 0.2 and 0.8 (Appendix A). Moreover, the comparison of the usual portion size consumed in one week (7-day recall, estimated with the FFQ) with the mean portion size value of the three 24 h recalls gave correlation coefficient values of at least 0.4 for all dishes, except for Ataclè and Abla. This was due to the limited number of respondents that did not allow for establishing a statistical relation. Moreover, the concordance percentages between the 24 h recall and the recorded weights ranged between 74 and 128%, except for Vêyi (162%), because it was consumed with other foods. The concordance percentages between FFQ and 24 h recall data varied between 66 and 96%, based on the Bland–Altman method.

### 3.2. Socio-Economic and Demographic Characteristics of FFQ Respondents

The respondents’ mean age was 39 years in Cotonou and 36 years in the two rural areas (Table 1). As data were weighted for the statistical analysis, we observed some small (not significant) differences between the number of interviewed women and men, which normally should have been the same in both area types. The education level was significantly higher in Cotonou than in the two rural areas, where almost half of participants never went to school and only 2% of respondents had a higher education diploma.

### 3.3. Consumption of Cowpea-Based Dishes

Nearly all respondents (98%) reported consuming cowpea-based dishes regularly and more than 70% at least once per week (Table 2). Between 90 and 95% of all respondents said that they had consumed at least one cowpea-based dish during the 7-day recall.

The mixed-dish Atassi and the stew Abobo were the most consumed cowpea-based foods in all study areas (Table 3). We observed some variations in the cowpea-based dish consumption patterns in the function of the area. Specifically, we found significant differences in the number of consumers between urban and rural areas, except for Atassi and Ataclè. Ata, Vêyi, and Adowè were more consumed by respondents in the urban area. Conversely, the number of consumers of Ata-doco, Abla and Djongoli in the rural areas was twice as much compared to the urban area. Similarly, the percentage of Abobo consumers was 15% higher in the two rural areas than in the urban area.

By grouping the different cowpea-based dishes consumed by each respondent during the 7-day recall, the total number of combinations reached 102 in the rural areas and 113 in the urban area. The most consumed combinations were a two-dish combination that included Abobo and Atassi (18% of all consumers) in the rural areas and a three-dish combination (Ata, Abobo, and Atassi; 11% of all consumers) in the urban area.

### 3.4. Consumption Frequency and Intake of Cowpea-Based Dishes

Among people who said they had consumed cowpea-based dishes during the 7-day recall, the frequency of consumption was significantly different between urban and rural consumers, except for Atassi, Abobo and Djongoli (Table 3). The frequency varied between 1.4 and 3.1 times per week, depending on the dish. However, when we took into account the people who said they did not consume these dishes during the 7-day recall, the mean consumption frequency decreased to 0.1–1.5 times per week for most dishes, except for Abobo and Atassi that were consumed 1.9–2.4 times per week. Based on these mean frequency values for each dish, we estimated the mean consumption frequency of cowpea-based dishes, all dishes included, at 7 times per week (7.2 ± 4.9 and 7.3 ± 5.4 in rural and urban areas, respectively): approximately once per day.

The mean portion size varied from one dish to another. Abobo, Atassi and Djongoli were consumed in quite large quantities (272 to 311 g/meal) (Table 3). Doughnuts (Ata, Ata-doco and Ataclè) and the stew Adowè were consumed in smaller portions. Among all survey participants, the mean intake per day of Adowè, Ata, Ata-doco, and Vêyi was significantly higher, and the mean intake per day for Atassi and Djongoli was lower in the urban area than in the two rural areas.

When we converted the daily intake of cowpea-based dishes into ‘cowpea seed equivalent’ (Table 3), the mean total quantity of cowpea seed equivalent consumed per day was 71 g in Cotonou and 58 g in the two rural areas (*p* < 0.05). This conversion into ‘cowpea seed equivalent’ also allowed us to calculate that, with cowpea consumption alone, 51.5% and 49.7% of the surveyed population, in urban and rural areas, respectively, reached the recommendations of the EAT-Lancet Commission on Healthy Diets from sustainable food systems [31] in terms of pulse consumption (i.e., at least 50 g pulses /day).

We observed some associations between socio-economic and demographic factors and (i) the daily intake of the studied dishes and (ii) the total quantity of cowpea seed equivalent consumed (Appendix A) in both area types. In rural areas, the daily intake (in cowpea seed equivalents) in Allada (54 g), even if substantial, was lower than in Adjohoun (67 g), in agreement with the local level of production (Adjohoun being an important cowpea production area). In addition, Appendix A shows that, in rural areas, the source of supply influences cowpea consumption: people who can buy directly from farmers have higher consumption than those who have to buy from the market. This influence of the supply source was not observed in urban areas. The total daily amount of cowpea seed equivalents consumed was significantly lower in women than in men (Appendix A), due to smaller portion sizes. In the rural areas, the consumption of cowpea seed equivalents per day was lower in the high compared to the low and middle socio-economic index classes. Moreover, in Cotonou, the quantity of cowpea seed equivalent consumed per day was lower in respondents with a higher education level than in the other education level groups (Appendix A). 

### 3.5. Nutritional Composition of Traditional Cowpea-Based Dishes

Table 4 presents the mean proximate composition, mineral, folate, and thiamine contents of the traditional cowpea-based dishes, as consumed. We observed high and significant variations (*p* < 0.05) among dishes for all the quantified nutrients: protein (3.0–9.9 g/100 g), dietary fibre (2.7–7.2 g/100 g), lipid (0.4–33.3 g/100 g), magnesium (13–68 mg/100 g), potassium (90–500 mg/100 g), zinc (0.3–1.6 mg/100 g), iron (0.4–8 mg/100 g), thiamine (28–175 µg/100 g) and folate (12–84 µg/100 g). Moreover, for each dish, we detected significant differences for most nutrients in function of the street food vendor (Appendix A). 

The doughnuts had a high lipid content, and consequently higher energy values, compared to the other dishes. The nutritional density of the three stews was higher than that of the other cowpea-based dish groups (Appendix A).

### 3.6. Contribution of Cowpea-Based Dishes to the RNI

The mean contribution of all cowpea-based dishes to the RNI of the studied nutrients (except iron, calcium, and thiamine) was higher than that for the energy. The consumption of the cowpea-based dishes during the 7-day recall allowed for the covering 42% of the RNI of dietary fibre, the highest percentage among the studied nutrients. Their contribution to the RNI of magnesium (37%), folate (30%), protein (26%), zinc (18%) and potassium (17%) was also interesting, being higher than 15% [32]. Conversely, their contribution to the RNI of calcium, iron, and thiamine was lower than 15% (Figure 3). 

Independently of the area (rural vs. urban), the Abobo stew, widely consumed and in large quantities by the adult population according to the 7-day recall data, was the dish that most contributed to the RNI of the studied nutrients (Appendix A). 

We observed that the number of different dishes consumed was strongly and positively associated with the amount of cowpea seed equivalent consumed (the correlation coefficient R was 0.64 and 0.81 for rural and urban areas, respectively). Consequently, the contribution to the RNI also increased with the number of different dishes consumed (Appendix A).

## 4. Discussion

In this study, we developed a FFQ to identify the consumption patterns of cowpea-based dishes and estimate their contribution to the RNI in adults in Cotonou and in two rural areas in the south of Benin. First, we performed two validation studies to assess the accuracy in the estimation of the quantity of the traditional cowpea-based dishes consumed, using a food atlas and a FFQ as tools. For this we used correlation tests and the Bland–Altman method that are commonly employed in validation studies at the individual and group level, respectively [33,34,35]. The correlation coefficients ranged between 0.2 and 0.8. Lombard et al. [36] showed that with correlation coefficients from 0.2 to 0.49, the relation between data is considered acceptable, and above 0.5 is considered good. Moreover, based on the results obtained using the Bland–Altman method, we can consider that the assessment of the usual intake with the FFQ can generate accurate data [27].

In our study, more than 70% of respondents said to consume cowpea-based dishes at least once per week in both urban and rural areas. This percentage was higher than that reported in 2009, in the survey (three non-consecutive 24 h food recalls) carried out among 200 urban adults in Benin [37], in which 56% of respondents were estimated to be cowpea consumers. It shows an increase in the number of consumers in the last decade instead of the decrease that was rather expected due to the nutritional transition in urban areas.

Among the cowpea-based dishes, the higher consumption of the stew Abobo and the mixed-dish Atassi could be explained by the fact that their processing involves few steps and that they can be easily prepared at home. Many other cowpea-based dishes are often bought as street food and consumed out of the house, particularly in urban areas where street food consumption is more developed than in rural areas [19]. This latter reason could explain the higher consumption frequency of Ata, Abobo, Vêyi and Adowè in urban areas than in rural areas. The consumption frequencies observed in our study were comparable with the results by Madodé et al. [15], showing that the current cowpea consumption frequency had not changed much compared with a decade ago. Most respondents declared consuming cowpea-based dishes at least once per week (Table 2). 

The total daily intake, expressed in cowpea seed equivalent, estimated at 71 g in the urban and 58 g in the rural areas, was in agreement with the value (77 g) reported by Delisle et al. [13] for legume consumption in south Benin. This corresponded to 49.7% of the population in rural areas and 51.5% in urban areas who consumed at least 50 g of pulses per day, as recommended by the EAT-Lancet Commission on healthy diets from sustainable food systems [31]. However, the chance to meet this recommendation was significantly lower for women living in the urban area (Appendix A). This observed consumption was also much higher than the 24.7 g for Benin in the country nutrition profiles by the Global Nutrition Report [38]. This important discrepancy could be partially due to consumption differences between the south and the other regions of Benin. Using the daily cowpea intakes of our study, the cowpea amount consumed could be estimated at 26 and 21 kg/adult/year in the urban and rural areas, respectively. This annual consumption is quite higher than in previous studies. Langyintuo et al. [39] reported that cowpea consumption was 9 kg/year/capita in Benin between 1990 and 1999, suggesting a doubling of the quantity consumed in the last three decades. 

The nutritional composition of traditional cowpea-based dishes varied widely depending on the street food vendors, reflecting variations due to cowpea cultivars, as concluded by Madode et al. [40], and processor practices. A previous study highlighted the influence of processors’ practices on the nutritional composition variability of cowpea-based doughnuts [14]. Despite these observed differences in composition, the high nutritional contribution, above 30% of the RNI of dietary fibre, magnesium and folates, showed that traditional cowpea-based dishes are rich in these compounds/nutrients. They can also be considered as a source (above 15% of the RNI) of protein, zinc and potassium. However, their contributions to the RNI of calcium and iron are low. It should be noticed that the reference values used to estimate the contribution to the recommended iron and zinc intake were those for low bioavailability, due to the presence of some mineral-chelating factors that reduce bioavailability (e.g., phytate) in cowpeas. Our findings raise questions about considering cowpea-based dishes as good sources of iron and calcium [8]. These results show that the consumption of cowpea-based dishes in southern Benin contributes to the nutritional security of the adult population, and undoubtedly helps reach adequate protein intakes. The very high daily contribution (above 40%) of the consumed cowpea-based dishes to the dietary fibre requirements shows that cowpea consumption can play a positive role in the prevention of overweight and obesity, and some dietary-related chronic diseases [11,41,42]. In addition, the high contribution to folate requirements may help prevent folate deficiency, which affects healthy foetal development in women of childbearing age, and is a public health problem in many countries around the world.

Our results finally show a positive relationship between the number of different cowpea-based dishes consumed and the total seed equivalent consumption. Moreover, we observed that for both urban and rural areas, the higher the number of different cowpea-based dishes consumed per week, the higher the overall contribution to the RNI. Thus, the availability of a wide variety of traditional cowpea-based dishes is also a factor that promotes total cowpea consumption: one cowpea-based dish not only replaces another cowpea-based dish, but also replaces other types of dishes.

Our study presents some strengths and limits. 

### 4.1. Strengths

The FFQ was validated for the estimation of the consumed quantities and consumption frequency of the nine traditional cowpea-based dishes.

The nutritional composition of the nine cowpea-based dishes used for the study were determined by laboratory analyses on samples collected in the study areas, and therefore reflected the actual nutritional value of these composite dishes. The resulting data sets are made available and may be used to complete food composition tables.

### 4.2. Limitations

Food intake estimation using a FFQ might lead to the underestimation or overestimation due to social desirability, as observed by Steinemann et al. [43]. Several studies reported overestimation when intakes were estimated with a FFQ compared with food records, especially for foods known to be beneficial for health, such as legumes, vegetables and fruits [44,45]. Therefore, some respondents might have overreported their usual consumed portion size or consumption frequency. Moreover, our surveys were focused on foods based on the same raw material and this could have led to an overestimation of the total cowpea seeds consumed per day due to a cumulative effect.

The surveys in Cotonou and in the two rural sites were separated by a few months, which could have led to seasonal variability. However, cowpea seeds are available all-year-round for purchase, and when directly questioned, respondents did not indicate any variation in consumption during the year.

## 5. Conclusions

Cowpea was highly consumed in the form of various dish types in all the study areas, resulting in important daily intakes. Indeed, with cowpea consumption alone, 51.5 and 49.7% of the surveyed population in urban and rural areas, respectively, reached the recommendations of the EAT-Lancet Commission on Healthy Diets from sustainable food systems, in terms of pulse consumption. The current consumption level by the adult population in the study areas corresponded to an estimated contribution by traditional cowpea-based dishes, all dishes combined, to the RNI of ~15% for energy, above 20% for protein, magnesium and folate, and over 40% for dietary fibre. The amount of cowpea seed equivalents consumed, and hence the contribution to nutritional requirements, was higher in participants who consumed a greater number of different cowpea dishes. This shows the value of having a wide variety of dishes. Thus, this regular and high cowpea consumption, together with that of other pulses, should be encouraged and maintained, as it may contribute to the nutritional security of the population, and may help prevent malnutrition.

## Figures and Tables

**Figure 1 nutrients-15-01314-f001:**
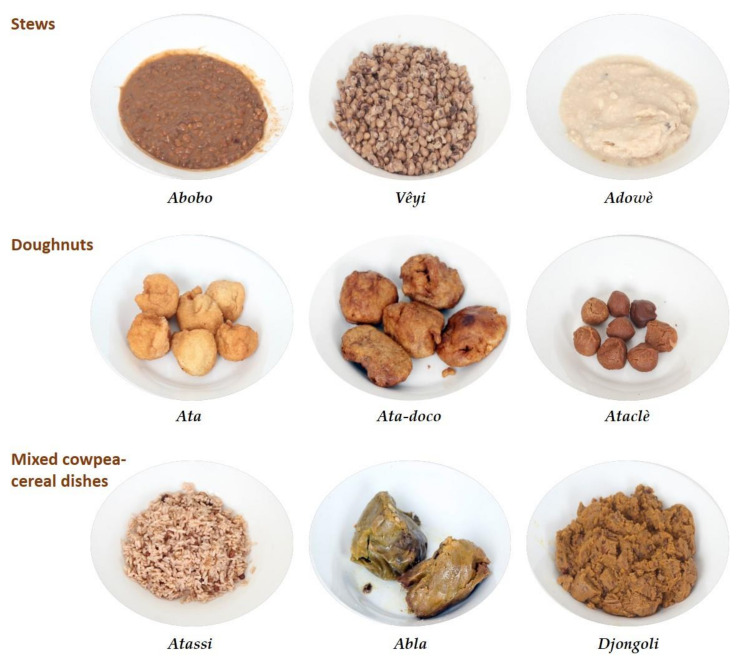
The nine traditional cowpea-based dishes consumed in Benin sampled in this study.

**Figure 2 nutrients-15-01314-f002:**
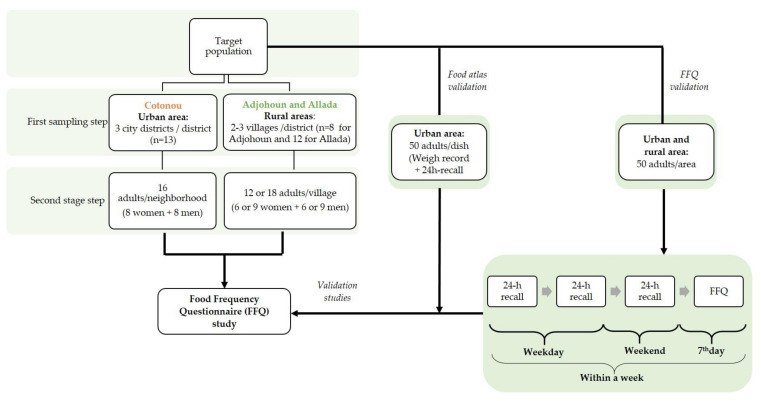
Schematic representation of the survey design.

**Figure 3 nutrients-15-01314-f003:**
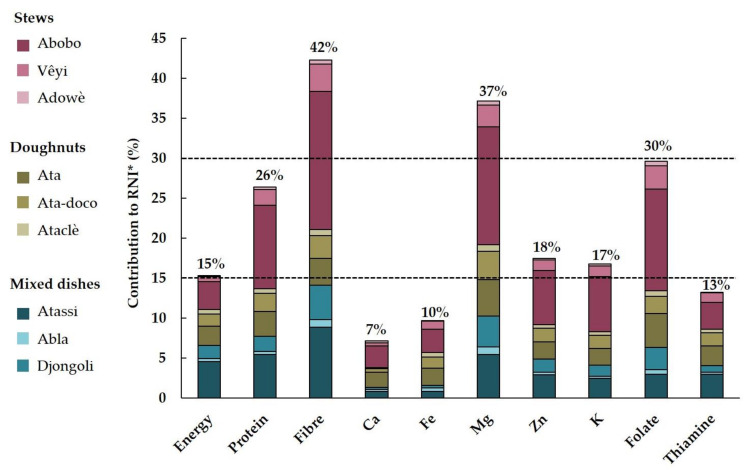
Mean total daily contribution to the RNI of the nine traditional cowpea-based dishes consumed during the week before the FFQ *. * All respondents, excluding non-consumers of cowpea seeds (i.e., 98% of respondents). Dotted lines show the thresholds to qualify a given food as source (15%) of, or as rich (30%) in protein, vitamins and minerals, as defined by the Codex Alimentarius (1997).

**Table 1 nutrients-15-01314-t001:** Socio-economic status and characteristics of the participants.

Characteristics	Urban Area (*n* = 641)		Rural Area (*n* = 576)		*p*-Value
	%	n	%	n	
Age					
<27 years	27	171	18	105	NS ***
27–34 years	26	169	21	119	
35–44 years	22	139	25	145	
45–65 years	25	162	36	207	
Mean ± SE	39.2 ± 0.7		35.7 ± 0.6		
Sex					
Men	51	324	49	285	NS *
Women	49	317	51	291	
SEI urban area (SEI 1)					
Low [0.4 ≤ Score < 1.2] ^1^	31	200	-		NA
Middle [1.2 ≤ Score < 1.8]	43	276	-		
High [1.8 ≤ Score < 3.0]	26	166	-		
SEI rural area (SEI 2)					
Low [0.1 ≤ Score < 1.3]	-		34	198	NA
Middle [1.3 ≤ Score < 1.8]	-		35	202	
High [1.8 ≤ Score < 3.1]	-		31	176	
Education					
None	16	101	45	256	<0.001 *
Primary school	28	177	29	168	
Secondary school	36	229	24	138	
Higher education	21	133	2	14	

NA: Not applicable; NS: Not significant; SE: Standard error; SEI: Socio-economic index. Values were obtained using weighted data. %: Percentage. n: sample size per group. ^1^: Values between [ ] are the SEI scores and correspond to the subjects’ coordinates on the first axis of the multiple-correspondence analysis. *: χ^2^ test between urban and rural areas.

**Table 2 nutrients-15-01314-t002:** Percentages of participants who consumed cowpea-based dishes and the frequency of consumption in the urban area (Cotonou) and the rural areas.

	Urban (*n* = 641)		Rural (*n* = 576)		*p*-Value ^1^
	%	*n*	%	*n*	
Consumption					NS
Usually	98	626 ^§^	98	565 ^§^	
Never consumed	2	15	2	11	
Frequency of consumption					<0.001
Never consumed	2	15	2	11	
<1 time/month	8	51	6	35	
1–3 times/month	5	32	22	127	
≥1 time/week	85	543	70	403	
7-day recall					0.005
Consumption during the last week	90	578 ^§§^	95	545 ^§§^	

^§^: Number of cowpea consumers. ^§§^: Number of consumers during the FFQ. ^1^: χ^2^ test for differences between urban and rural areas.

**Table 3 nutrients-15-01314-t003:** Quantities of cowpea-based dishes consumed (g, fresh weight) and cowpea seed equivalent (g, fresh weight) in the urban and rural study areas.

		Number of Consumers in the Week before the FFQ *			Consumption Frequency (Times/Week) in the Week before the FFQ ^ǂ^			Usual Portion Size per Meal (g) ^ǂ^			Intake (g of Dish/Day) ^ǂ^			Cowpea Seed Equivalent *** (g, per Day) ^ǂ^		
		Urban	Rural	*p*-Value ^1^	Urban	Rural	*p*-Value ^2^	Urban	Rural	*p*-Value ^2^	Urban	Rural	*p*-Value ^2^	Urban	Rural	*p*-Value ^2^
Participants who consumed the dish in the week before the FFQ-based survey																
Stews	Abobo	441	496	<0.001	2.8	2.6	NS	299	273	0.02	121	106	0.02			
	Vêyi	142	77	<0.001	2.5	1.8	<0.001	236	259	NS	87	66	0.005			
	Adowè	150	35	<0.001	2.4	1.7	0.008	65	80	NS	23	21	NS			
Doughnuts	Ata	350	122	<0.001	2.7	2.0	<0.001	162	117	<0.001	65	34	<0.001			
	Ata-doco	92	188	<0.001	2.8	2.0	<0.001	292	98	<0.001	120	30	<0.001			
	Ataclè	38	35	NS	2.4	1.4	<0.001	211	103	0.001	59	23	0.004			
Mixed dishes	Atassi	444	440	NS	2.9	3.1	NS	297	305	NS	127	137	NS			
	Abla	30	85	<0.001	2.3	1.5	0.002	231	216	NS	92	48	0.05			
	Djongoli	95	213	<0.001	2.0	2.2	NS	284	311	0.04	82	99	0.02			
All participants (including those who did not consume the dish in the week before the FFQ) **																
Stews	Abobo	441	496	<0.001	1.9	2.3	<0.001	86	93	NS	27	29	NS	26.9	29.3	NS
	Vêyi	142	77	<0.001	0.6	0.2	<0.001	20	9	<0.001	7	3	<0.001	7.4	3.4	<0.001
	Adowè	150	35	<0.001	0.6	0.1	<0.001	6	1	<0.001	2	0	<0.001	1.7	0.4	<0.001
Doughnuts	Ata	350	122	<0.001	1.5	0.4	<0.001	37	7	<0.001	14	3	<0.001	14.4	2.9	<0.001
	Ata-doco	92	188	<0.001	0.4	0.7	0.003	18	10	0.02	8	4	0.02	7.7	4.4	0.02
	Ataclè	38	35	NS	0.1	0.1	NS	4	1	NS	2	1	0.04	2.3	0.7	0.04
Mixed dishes	Atassi	444	440	NS	2.1	2.4	0.008	90	106	0.01	8	9	0.01	7.8	9.2	0.01
	Abla	30	85	<0.001	0.1	0.2	0.01	4	7	NS	1	1	NS	0.5	0.9	NS
	Djongoli	95	213	<0.001	0.3	0.8	<0.001	12	37	<0.001	2	7	<0.001	2.2	6.5	<0.001

All values were obtained using weighted data. *: The total number of cowpea consumers during the FFQ week was *n* = 578 for urban area and *n* = 545 for rural areas. ^1^: χ^2^ test for differences between urban and rural areas; ^2^: One factor analysis of variance. **: All respondents in each area, excluding non-consumers of cowpea seeds. ǂ: Values are the mean. NS: Not significant. ***: Quantity calculated by considering the proportion of cowpea in the dish and the processing yield.

**Table 4 nutrients-15-01314-t004:** Nutritional composition of the nine cowpea-based dishes (per 100 g of edible portion).

	Dishes	Water (g)	Protein (g)	Lipid (g)	Available Carbohydrate (g)	Dietary Fibre (g)	Ash (g)	Energy (kcal)	Fe (mg)	Zn (mg)	Ca (mg)	Mg (mg)	K (mg)	Folate (µg)	Thiamine (µg)
Stews	Abobo	71.6 ± 2.9	6.3 ± 0.7	0.5 ± 0.0	14.3 ± 2.6	5.8 ± 0.2	1.5 ± 0.4	98± 13	1.4 ± 0.3	0.9 ± 0.2	30.3 ± 5.8	39 ± 7.8	272 ± 51	56.2 ± 4	43.5 ± 13
	Vêyi	66.1 ± 2.6	7.5 ± 0.4	0.6 ± 0.1	16.7 ± 2.0	7.2 ± 0.4	1.9 ± 0.3	116 ± 9	1.5 ± 0.1	1.1 ± 0.1	27.9 ± 4.8	46 ± 3.6	323 ± 39	80.5 ± 7	89.6 ± 42
	Adowè	73.2 ± 3.4	6.3 ± 1.2	0.4 ± 0.2	13.7 ± 2.5	5.0 ± 0.3	1.5 ± 0.1	94 ± 14	1.1 ± 0.2	0.9 ± 0.2	25.7 ± 7.5	36 ± 4.2	258 ± 37	65.1 ± 22	28.4 ± 21
Doughnuts	Ata	47.6 ± 5.7	7.4 ± 0.5	19.5 ± 7.5	18.7 ± 1.7	4.7 ± 0.3	2.0 ± 0.2	289 ± 61	4.2 ± 0.4	1.2 ± 0.1	28.2 ± 7.6	50 ± 1.4	333 ± 19	74.3 ± 19	133.3 ± 18
	Ata doco	44.3 ± 5.9	8.8 ± 1.0	16.5 ± 7.2	21.9 ± 1.1	6.2 ± 0.4	2.2 ± 0.4	284 ± 60	4.1 ± 1.2	1.5 ± 0.1	27.9 ± 4.8	61 ± 5.2	425 ± 56	64.0 ± 18	135.8 ± 26
	Ataclè	23.9 ± 2.6	9.9 ± 0.3	33.3 ± 3.8	23.9 ± 1.6	7.0 ± 0.5	2.2 ± 0.2	449 ± 29	8.0 ± 6.8	1.6 ± 0.2	33.4 ± 7.3	68 ± 5.6	500 ± 31	84.3 ± 22	174.9 ± 23
Mixed dishes	Atassi	68.9 ± 1.3	3.0 ± 0.1	0.7 ± 0.4	23.7 ± 0.8	2.7 ± 0.2	1.0 ± 0.1	119 ± 7	0.4 ± 0.1	0.3 ± 0.1	8.2 ± 2.2	13 ± 4.2	90 ± 27	12.2 ± 3.2	34.3 ± 5
	Abla	66.1 ± 3.5	3.6 ± 0.9	10.2 ± 1.5	13.6 ± 2.5	5.1 ± 0.4	1.3 ± 0.3	171 ± 17	3.0 ± 1.6	0.7 ± 0.1	25.0 ± 2.1	39 ± 1.4	167 ± 51	40.3 ± 17	50.3 ± 7
	Djongoli	63.5 ± 5.3	4.2 ± 1.2	7.2 ± 2.6	18.6 ± 2.5	5.2 ± 0.6	1.3 ± 0.1	166 ± 34	1.5 ± 0.3	0.8 ± 0.1	9.8 ± 2.2	37 ± 6.7	193 ± 51	48.0 ± 39	44.3 ± 9

Values are the mean ± standard deviation, *n* = 3 (*n*: number of street food vendors).

## Data Availability

Data presented in this study are available in the following datasets: Akissoé, L.; Hemery, Y.; Icard Verniere, C.; Rochette, I.; Picq, C.; Devaux, S.; Cros, L.; Mouquet-Rivier, C., 2023, “Nutritional composition of traditional cowpea-based dishes sampled in South Benin”, https://doi.org/10.23708/MQLMLJ DataSuds, V1; and Akissoé, L.; Hemery, Y.; Madode, Y.; Icard Verniere, C.; Roger, A.; Kpossilande, C.; Hounhouigan, D.J.; Mouquet-Rivier, C., 2023, “Consumption of traditional cowpea-based dishes in urban and rural areas of South Benin”, https://doi.org/10.23708/ZQPBEW, DataSuds, V1.

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
