# Peer review of "Current Consumption of Traditional Cowpea-Based Dishes in South Benin Contributes to at Least 30% of the Recommended Intake of Dietary Fibre, Folate, and Magnesium"

_nutrients, 2023, doi:10.3390/nu15061314_

Round 1
Reviewer 1 Report
The present paper describes the consumption of cowpea based dishes in different regions of Benin and their contribution to the recommended dietary intakes. The study is adequately designed and the article is written clearly. Some comments to the author to improve the manuscript:
1. the authors mention 2 rural regions where cowpea is produced in different amounts. The data is then pooled for both regions, but it would be interesting to see if the availability/access is among the reasons for the observed differences in consumption between rural and urban regions. The influence of education could also be discussed since there is data available.
2. The number of responders should be given for the validation of food atlas together with the p values. The Spearman correlation koeficient has no value without the calculation of significance.
3. The numbers in the text regarding table 4 should be controlled as there are some inconsistencies with the numbers in the table. The term fats and lipids should not be used interchangeably.
4. In section 3.2 the "standard methods" should be stated by name
5. In figure 2 6/9 should be replaced with 6 or 9, in table 3 first column: consumed what?
Reviewer 2 Report
The authors studied the contribution of nine cowpea-based foods to the adult RNIs of dietary nutrients in southern Benin using questionnaire and nutritional composition data. This study did not provide much novel findings as the consumption of cowpea dishes in Benin or Western Africa were reported in previous studies. I have some comments:
1. Did the authors have data on the BMI of the respondents? Are they all healthy or having any metabolic disorders including obesity, diabetes and high blood pressure? This would be important and the novel point to determine whether there are correlations between the consumption pattern cowpea-based dishes with the BMI and/or with metabolic disease incidence.
2. For the measurement of nutritional composition, only three samples were obtained for each type of dish and all were from urban area (Cotonou). The data were not conclusive as they might be different from those in rural areas.
3. The data presentation is not professional and difficult to read. I cannot find data showing consumption pattern correlate with age, gender, and area.
4. Overall, there is not sufficient and solid evidence to support the conclusion
Round 2
Reviewer 2 Report
the authors have addressed my concerns and comments. I have no more comments.